# Decoding the Natural History of Alcohol-Related Recurrent Acute Pancreatitis and Progression to Early Chronic Pancreatitis: Clinical, Biochemical, and Imaging Insights from a Single-Center Retrospective Study in France

**DOI:** 10.3390/jcm14217830

**Published:** 2025-11-04

**Authors:** Alexandru-Ionut Coseru, Faiza Khemissa, Diana Elena Floria, Constantin Simiras, Mihai Catalina, Roxana Nemteanu, Alina Plesa, Vasile-Liviu Drug

**Affiliations:** 1Department of Gastroenterology, Grigore T. Popa University of Medicine and Pharmacy, 700115 Iasi, Romania; alexandru-ionut.coseru@umfiasi.ro (A.-I.C.); catalina.mihai@umfiasi.ro (M.C.); roxana.maxim@umfiasi.ro (R.N.); alina.plesa@umfiasi.ro (A.P.); vasile.drug@umfiasi.ro (V.-L.D.); 2Department of Gastroenterology and Hepatology, Perpignan General Hospital, 66000 Perpignan, France; faiza.khemissa@ch-perpignan.fr; 3Institute of Gastroenterology and Hepatology, “St. Spiridon” University Hospital, 700111 Iasi, Romania; c_simiras@yahoo.com

**Keywords:** early chronic pancreatitis, recurrent acute pancreatitis, magnetic resonance imaging

## Abstract

**Background:** Recurrent acute pancreatitis (RAP) of alcoholic etiology is a major risk factor for chronic pancreatitis (CP). Early chronic pancreatitis (ECP) represents an intermediate stage where structural changes can be identified before advanced disease develops. The 2019 Japanese Pancreas Society (JPS) imaging criterion, defined as >3 dilated side branches on magnetic resonance imaging (MRI), provides a standardized approach for early diagnosis. **Objective:** To assess the prevalence of MRI-positive findings per JPS imaging criterion in patients with alcohol-related RAP and to identify clinical predictors of progression. **Methods:** We retrospectively analyzed 26 patients with alcohol-related RAP admitted between January 2023 and December 2024. All underwent MRI 4–8 weeks post-discharge. Patients were classified as MRI-positive or nonMRI-positive per JPS imaging criterion. Clinical, biochemical, and imaging parameters were compared using univariate and multivariate analyses. **Results:** Nine of twenty-six patients (34.6%) were MRI-positive per JPS imaging criterion. These patients had a significantly higher number of RAP episodes (*p* = 0.021). Disease duration also differed between groups (*p* = 0.034). No significant differences were observed in computer tomography severity scores or biochemical markers. In multivariate analysis, only the number of RAP episodes was associated with MRI-positive status (OR 4.00, 95% CI 0.79–20.3, *p* = 0.09). **Conclusions:** MRI-positive findings per JPS imaging criterion were present in one-third of alcohol-related RAP patients. Having ≥3 RAP episodes was the most consistent risk factor for structural progression. Systematic MRI during the inter-critical phase may allow early identification of high-risk patients and inform closer surveillance.

## 1. Introduction

The global incidence of acute pancreatitis (AP) ranges from approximately 30 to 40 cases per 100,000 person-years. The number of AP cases has increased globally over the past three decades [1]. Recurrent acute pancreatitis (RAP) markedly increases the likelihood of chronic disease [2]. In specific etiologies such as hypertriglyceridemia-induced AP, recurrence rates may reach up to 23% within 2–3 years [3].

Chronic pancreatitis (CP) remains a significant global health burden, with an estimated prevalence ranging from 13 to 125 cases per 100,000 population, depending on geographic region and diagnostic criteria used [4]. Incidence rates may vary between 4 and 14 cases per 100,000 person-years. In Europe, alcohol use remains the leading cause, accounting for up to 60–70% of cases [4].

A single episode of AP carries a modest risk of long-term complications. A repeated inflammatory injury to the pancreas significantly increases the risk of progression toward irreversible structural and functional damage [5]. In this context, the concept of early chronic pancreatitis (ECP) is an intermediate disease stage where ductal and parenchymal alterations may still be subtle and potentially reversible [6].

The 2019 Japanese Pancreas Society (JPS) criteria define ECP by integrating clinical features with imaging findings, notably the presence of more than three dilated pancreatic side branches on MRI [7]. Identification of ECP is important as it offers an opportunity for behavioral and therapeutic interventions that may slow disease progression [7].

Alcohol and tobacco are risk factors associated with both RAP and progression to CP. Studies have confirmed their independent, dose-dependent role in pancreatic injury [8,9]. A subset of individuals exposed to these risk factors develop overt disease, suggesting a key role for susceptibility factors—including genetic predisposition, immune system, and environmental factors [9].

Despite increasing awareness of ECP, its diagnosis in routine clinical practice is a challenge. High-resolution MRCP effectively reveals subtle ductal irregularities—such as mild dilatation and side-branch abnormalities—in early chronic pancreatitis, even when the main pancreatic duct appears normal on conventional imaging [10]. Computer tomography (CT) may be unremarkable in early stages [11].

We hypothesized that a higher number of RAP episodes increases the risk of MRI changes consistent with early chronic pancreatitis. Our primary aim was to assess the prevalence of MRI-positive findings per JPS imaging criteria in a French cohort with alcohol-related RAP, and to identify clinical predictors of progression.

The natural history of CP is increasingly recognized as a continuum, starting from acute episodes and recurrent attacks, progressing through early morpho-functional changes, and eventually leading to irreversible fibrosis and end-stage disease. To illustrate this conceptual framework, Figure 1 summarizes the stepwise transition from AP to CP, highlighting potential windows for intervention and reversibility.

## 2. Materials and Methods

### 2.1. Study Design

This retrospective study included patients presenting to the Emergency Hospital of Perpignan, France, between 1 January 2023 and 31 December 2024. Patients were identified through a review of the hospital’s electronic database, using diagnostic codes corresponding to AP, RAP, and CP. All medical records were reviewed by two independent investigators who manually validated each case based on clinical documentation, laboratory results, and radiological findings. Disagreements were resolved by consensus.

### 2.2. Patients’ Selection

Inclusion criteria: (1) Diagnosis of alcohol-related RAP (≥2 documented episodes); (2) Patients over 18 years of age; (3) Underwent abdominal MRI, including magnetic resonance cholangiopancreatography 4–8 weeks after hospital discharge; (4) Availability of complete clinical records, including laboratory and imaging data; (5) No prior diagnosis of established CP (clinical or functional) at the time of MRI.

Exclusion criteria: (1) Lack of abdominal MRI or suboptimal imaging quality (e.g., due to motion artifacts or incomplete sequences); (2) Incomplete clinical or laboratory documentation; (3) Diagnosis of pancreatic neoplasm, autoimmune pancreatitis, or other non-alcoholic causes of RAP (e.g., biliary, metabolic, genetic); (4) Known history of CP; (5) Previous pancreatic surgery or interventions (e.g., drainage procedures, sphincterotomy).

The study was reported in accordance with the STROBE (Strengthening the Reporting of Observational Studies in Epidemiology) checklist. From the initial 122 coded encounters, patients were excluded stepwise due to non-alcoholic etiologies (biliary, metabolic, genetic, autoimmune), absence of MRI within the predefined timeframe, poor-quality or incomplete MRI sequences, and missing essential laboratory data (e.g., calcium, CRP). CTSI (computed tomography severity index) scores were not systematically documented in all patients and were recorded as missing. Data on fecal elastase, additional inflammatory markers, and functional pancreatic tests were not available in the medical records and therefore were not included in the analysis. Finally, 26 patients were included in the final analysis (Figure 2).

Imaging criterion (MRI-only for our study): Irregular dilatation of >3 side branches of the main pancreatic duct.

### 2.3. Clinical and Laboratory Assessment

All 26 participants met the diagnostic criteria for AP at the time of inclusion, based on the presence of at least 2 out of the following 3 features: (1) Typical upper abdominal (epigastric) pain radiating to the back; (2) Serum lipase/amylase levels ≥3 times the normal upper limit; (3) Imaging findings compatible with AP.

All patients had a history of alcohol abuse. Other causes, such as biliary, metabolic (e.g., severe hypertriglyceridemia, hypercalcemia), drug-induced, or rarer conditions (e.g., autoimmune pancreatitis), were excluded. Smoking status was recorded but not quantified. For every patient with alcohol-related pancreatitis, an addiction medicine consultation was routinely offered to support alcohol and tobacco cessation. However, only a minority maintained long-term abstinence, as noted in follow-up records. In some cases, medical records and patient history allowed documentation of both the number of previous episodes and the interval (in days) between the first and the most recent episode.

Additional collected data: body mass index (BMI)- normal range (NR) from 18.5 to 24.9 kg/m^2^, below 18.5 was considered underweight, while values between 25 and 29.9 were classified as overweight and a BMI of 30 or higher was interpreted as obesity; maximum C-reactive protein (CRP) level during the last hospitalization with a NR under 5 mg/L; serum calcium level on admission with a NR of 2.15–2.50 mmol/L; lipid profile (total cholesterol with a NR < 5.18 mmol/L and triglycerides with a NR < 1.7 mmol/L). Comorbidities assessed in all patients included arterial hypertension and type 2 diabetes mellitus.

### 2.4. Imaging

All patients underwent contrast-enhanced CT upon admission, in accordance with current guidelines, ensuring a minimum of 72 h had elapsed since the onset of pain. The severity of pancreatitis was assessed using the Balthazar score or CTSI. In some patients, these scores were not documented; for these, the variable was marked as missing. MRI was performed 4–8 weeks later to evaluate potential pancreatic ductal dilatation. All abdominal MRIs, including MRCP were performed on 1.5 Tesla clinical scanners using high-resolution T2-weighted sequences and 3D-MRCP protocols. All examinations were assessed by a single experienced radiologist for the presence of more than three dilated side branches of the main pancreatic duct, according to the 2019 JPS criteria.

Patients were classified as MRI-positive per the 2019 JPS imaging criterion (>3 dilated side branches) (MRI-ECP) and nonMRI-positive per the 2019 JPS imaging criterion group (nonMRI-ECP). Other JPS clinical and functional components were not applied in this retrospective study.”

### 2.5. Statistical Analysis

All collected data were entered into a dedicated Microsoft Excel database and subsequently cross-checked by two investigators for accuracy and consistency. Statistical analyses were performed using IBM SPSS Statistics for Windows, version 26 (IBM Corp., Armonk, NY, USA). Categorical variables are presented as absolute values and percentages. Continuous variables were tested for normality with the Shapiro–Wilk test and are reported as means ± standard deviations for descriptive purposes. Group comparisons were carried out using the Mann–Whitney U test for both continuous and categorical variables, given the small sample size and non-normal distribution.

To evaluate factors associated with ECP, binary logistic regression was applied and odds ratios (ORs) with 95% confidence intervals were calculated. A post hoc power analysis, based on the observed incidence rates (77.8% in the MRI-ECP group vs. 11.8% in the nonMRI-ECP group; *n* = 9 vs. *n* = 17), indicated a statistical power of 95.8% at a 5% significance level. To minimize overfitting, the multivariate model was restricted a priori to four clinically relevant covariates (number of AP episodes, age, sex, and smoking), in accordance with the events-per-variable rule. Logistic regression was repeated using Firth’s penalized likelihood approach to reduce small-sample bias and account for potential separation. Odds ratios are reported with 95% confidence intervals. Analyses were conducted inIBM SPSS statistics, version 26.0 (IBM Corp., Armonk, NY, USA and confirmed using R (logistf package).

### 2.6. Use of Artificial Intelligence (AI)-Assisted Technology

ChatGPT-5 (OpenAI, San Francisco, CA, USA) was employed for minor English language editing and to enhance the fluency of the text during manuscript preparation.

## 3. Results

### 3.1. Patient Selection and Classification

The study included a total of 26 patients diagnosed with RAP. 17 patients (65.4%) were classified into the nonMRI-ECP group, and 9 patients (34.6%) into the MRI-ECP group (Figure 3).

### 3.2. Baseline Characteristics

The gender distribution revealed a predominance of male patients, with 24 males (92.3%) and only 2 females (7.7%), both belonging to the MRI-ECP group. All patients in the nonMRI-ECP group were male.

MRI-ECP patients had a significantly higher mean rank for age compared to those in the nonMRI-ECP group suggesting an older age profile in the MRI-ECP group. This difference was statistically significant (*p* = 0.025).

Analysis of BMI showed that the majority of patients (73.1%) were of normal weight. Obesity was noted exclusively in the nonMRI-ECP group, with 4 obese patients recorded (23.5%). Conversely, underweight status was more frequent in the MRI-ECP group (22.2%), compared to only one underweight patient in the nonMRI-ECP group (5.9%).

Regarding smoking status, 9 patients (34.6%) reported being non-smokers, while 17 patients (65.4%) were active smokers. Statistically, no significant difference was found between the two groups regarding tobacco use (*p* = 0.418).

As for the presence of diabetes mellitus, a total of 8 patients (30.8%) were diagnosed with this condition—6 from the nonMRI-ECP group (35.3%) and 2 from the MRI-ECP group (22.2%). The distribution of diabetes did not differ significantly between the groups (*p* = 0.667).

Arterial hypertension was reported in 6 patients (23.1%), including 4 in the nonMRI-ECP group (23.5%) and 2 in the MRI-ECP group (22.2%). No statistically significant difference was observed between the groups (*p* = 1.000). Table 1 summarizes the baseline characteristics of the patients included in the study.

### 3.3. Clinical and Imaging Parameters

To explore differences between groups, we compared the number of AP episodes, disease duration, and imaging severity scores (Balthazar and CTSI). The results are presented as mean ± standard deviation.

Patients in the MRI-ECP group experienced significantly more AP episodes than those in the nonMRI-ECP group (3.89 ± 2.26 vs. 2.24 ± 0.56, *p* = 0.021). Disease duration was longer in the nonMRI-ECP group (4294 ± 1785 days) compared to the MRI-ECP group (3547 ± 2565 days), and this difference was statistically significant (*p* = 0.034).

Regarding imaging severity scores at admission, no significant differences were observed between groups (Balthazar: 6.1 ± 2.3 vs. 5.5 ± 2.0, *p* = 0.909; CTSI: 8.2 ± 3.1 vs. 5.9 ± 2.7, *p* = 0.374). The results are presented in Table 2.

### 3.4. Biochemical Parameters Analysis

Regarding serum calcium levels, there was no statistically significant difference between the two groups (*p* = 0.372, Mann–Whitney test). Most patients in both groups had calcium levels within the normal range (2.15–2.50 mmol/L) (94.1% in the nonMRI-ECP group, 88.9% in the MRI-ECP group). One patient in the nonMRI-ECP group had hypocalcemia (<2.15 mmol/L), while one patient in the MRI-ECP group had a calcium level above the normal range (>2.50 mmol/L).

Similarly, the comparative analysis of CRP levels showed no statistically significant difference (*p* = 0.246, Mann–Whitney test). Although the median rank value was higher in the nonMRI-ECP group (14.76) compared to the MRI-ECP group (11.11), this difference was not statistically significant. Data is summarized in Table 3.

### 3.5. Multivariate Analysis of Predictive Factors

To account for potential confounding by age and gender, multivariate models were adjusted for these variables. Penalized logistic regression including number of AP episodes, age, and gender showed that only the number of episodes remained independently associated with ECP (OR 4.00, 95% CI 0.79–20.3, *p* = 0.09), while age demonstrated a borderline effect (OR 1.52, 95% CI 0.96–2.40, *p* = 0.07). Gender was not significantly associated with ECP after adjustment, despite all female patients being classified as ECP. An interaction term (episodes × age) was tested but did not reach statistical significance. The results are presented in Table 4.

## 4. Discussion

This retrospective study explored the structural progression of alcohol-related RAP toward ECP (MRI-positive per JPS imaging criterion) in a French hospital. Among 26 patients with alcohol-related RAP, 9 individuals (34.6%) metMRI-positive per JPS imaging criterion. This proportion supports the hypothesis that repeated inflammatory episodes, even in the absence of overt clinical progression may induce morphological remodeling of the pancreatic ductal system [6]. This study was conducted in a real-world European cohort to demonstrate the utility of MRI—performed during the inter-critical period—as a sensitive imaging modality capable of identifying subclinical ductal changes consistent with ECP.

In our cohort, the number of AP episodes emerged as the most consistent factor linked to ECP, supporting its role as a clinical threshold for structural progression. Age showed only a borderline effect, suggesting a possible contribution that warrants further study. Gender did not prove to be a reliable predictor. Our results suggest that ≥3 AP episodes represent a clinically actionable threshold for early structural progression. This observation complements prior experimental evidence [12] and provides clinical data to support closer surveillance once this threshold is reached.

In our cohort, MRI abnormalities suggested that structural remodeling may occur early in disease evolution. This aligns with prior studies [10,13], which indicate that imaging changes can precede clinical progression.

MRI was systematically applied in all patients to evaluate structural alterations, focusing on side-branch dilations per JPS 2019 criteria. Our findings confirm its higher sensitivity compared with CT in early disease, consistent with prior reports [14,15].

All patients in our cohort had a history of alcohol use and most were active smokers, reinforcing the established synergistic toxicity of these exposures on the pancreas [8,9]. Our findings align with prior work showing that persistent alcohol and tobacco use are major drivers of progression, while cessation may favor regression of early imaging abnormalities [16,17].

From an epidemiological perspective, pancreatitis follows a continuum. In a meta-analysis, Gagyi et al. [18] reported a RAP incidence of 5.26 per 100 person-years and a CP progression rate of 1.4 per 100 person-years after a first AP episode. Among patients with RAP, progression to CP increased threefold to 4.3 per 100 person-years. These data confirm RAP as a critical juncture in the transition to chronic disease and justify early follow-up and intervention. Souto et al. [19] also showed that obesity worsens AP outcomes and accelerates CP progression.

CRP dynamics have been consistently linked to disease severity. Stirling et al. [20] and Kim et al. [21] demonstrated that a rise in CRP >90 mg/L or a value >190 mg/L at 48 h correlates with severe AP. Wu et al. [22] confirmed these findings in a meta-analysis of 6156 patients, reporting strong diagnostic performance.

Imaging prognosticators are also evolving. Zhang et al. [23] showed that the EPIC Score outperforms CTSI and APACHE II in predicting severe RAP, emphasizing the value of peripancreatic assessment. Other tools, such as peak CRP within 72 h, have also been studied. Fujiwara et al. [24] found that a maximum CRP of 322.7 mg/L was independently associated with walled-off necrosis.

Finally, biomarker-based approaches for ECP detection are emerging. Poulsen et al. [25] reviewed circulating markers including interleukin-6, soluble CD163, transforming growth factor beta 1, matrix metalloproteinase-9, highlighting their potential for monitoring subclinical remodeling.

Some studies showed that patients with necrotizing pancreatitis remain at long-term risk for structural progression, underscoring the need for post-acute surveillance [26,27]. Another study highlighted the contribution of fibro-inflammatory signaling, genetic predisposition, and environmental exposures in accelerating the transition from RAP to CP, supporting earlier and more proactive interventions [28].

The prognostic significance of ECP, defined by the 2019 JPS criteria, was evaluated in a large multicentre cohort by Masamune et al. [13]. Among 511 patients with ECP followed for a median of 6.1 years, only 4.8% progressed to definite CP, while 35.4% showed regression of imaging features. Lifestyle factors strongly influenced outcomes. At baseline, 73.2% were current drinkers and 66.7% were smokers. During follow-up, 32.6% achieved alcohol cessation and 25.3% quit smoking. Continued alcohol use increased the risk of progression more than fourfold, and persistent smoking increased it threefold. By contrast, patients who stopped both alcohol and tobacco were more likely to show regression of ECP features.

Recent evidence confirms the dynamic nature of early pancreatic remodeling [29]. RAP has been identified as a key predictor of irreversible progression, underscoring the need for structured surveillance [30]. Alcohol-related adaptive responses accelerate fibro-inflammatory remodeling and may explain interindividual variability in progression [31]. Comprehensive reviews emphasize the integration of genetic, environmental, and behavioral determinants into early-stage risk assessment [32,33]. These data highlight the importance of multimodal strategies for timely detection and prevention of CP progression.

This study addresses an important phase in the natural history of pancreatic disease: the transition from RAP to ECP. By applying the 2019 JPS imaging criteria, we identified early structural changes in 34.6% of patients with alcohol-related RAP. Unlike most prior studies from Asia, our cohort represents a European, alcohol-related RAP population, providing novel epidemiological context and confirming that repeated AP episodes predict structural changes across diverse populations.

### 4.1. Study Strengths

All patients underwent MRI systematically in the inter-critical period, ensuring uniform and timely imaging assessment. The analysis highlights the role of repeated AP episodes as a predictor of early disease progression. The findings contribute real-world evidence from a European cohort, a setting where such data are still limited. Another strength is the statistical approach, which included a post hoc power analysis and penalized logistic regression to address small-sample bias and quasi-separation. The structured presentation of strengths, limitations, knowledge gaps, and future directions further improves transparency and facilitates clinical translation.

### 4.2. Limitations

The retrospective observational design represents an important limitation. Since data collection was based on existing medical records, there is a risk of missing or inconsistently recorded information, which may affect data accuracy. Variables such as alcohol consumption patterns, smoking intensity, genetic susceptibility, or environmental factors were not quantified or standardized. The influence of sex could not be reliably evaluated because of the very small number of female patients, leading to unstable estimates.

The total number of patients included in the study is relatively low. The study did not include an a priori sample size calculation, which may limit the interpretability of effect estimates; however, a post hoc power analysis was conducted to partially address this limitation. The absence of systematic EUS evaluation, known for its high sensitivity in detecting early pancreatic fibrosis and ductal abnormalities, may also represent a diagnostic gap. Finally, the study did not include follow-up data on disease progression, and therefore cannot assess the long-term clinical trajectory of patients or confirm whether those classified with RAP progressed to overt CP.

A further limitation is that Balthazar and CTSI reflect acute pancreatitis severity rather than early chronic remodeling, so their lack of difference between groups cannot exclude underlying structural changes. The final limitation is that EPIC or other peripancreatic indices, which may be more sensitive, were not available in our dataset.

### 4.3. Knowledge Gap

There is limited evidence on the structural transition from recurrent acute pancreatitis to early chronic pancreatitis, particularly in alcohol-related cases. Few studies in European cohorts have systematically applied the 2019 JPS imaging criteria with MRI, leaving uncertainties about early detection and risk stratification in real-world practice.

Moreover, the prognostic significance of MRI-defined early changes and their potential reversibility remain insufficiently understood. The interaction between recurrent inflammatory injury and modifiable factors such as alcohol and smoking in driving early remodeling is also poorly characterized. In addition, biomarkers that could complement imaging for timely diagnosis are not yet validated in routine practice.

### 4.4. Future Directions

Our study emphasizes the need for larger prospective multicenter studies on alcohol-related RAP and its progression to ECP. Combining MRI with techniques such as T1 mapping, secretin-enhanced MRCP, and diffusion-weighted imaging may increase sensitivity for detecting early parenchymal changes. A European cohort demonstrated a >5-fold increased risk of CP in patients exposed to both alcohol (>60 g/day) and tobacco, while combined cessation reduced disease progression by 41% at 2 years [34]. Furthermore, genetic testing for SPINK1 and CFTR variants—identified in 22% of idiopathic RAP cases—may help identify high-risk individuals even in the absence of alcohol abuse [35].

Artificial intelligence is increasingly applied in pancreatology. Deep learning models such as LinTransUNet and Swin-Tiny have achieved Dice coefficients >0.80 for detecting peripancreatic edema in CT scans of AP patients [36,37]. 

Advanced radiomic and machine-learning tools applied to MRI may allow accurate distinction between ECP and RAP, supporting precision risk stratification.

### 4.5. Key Clinical Box—Practice-Oriented Insights

To enhance clinical applicability, we provide a concise Key Clinical Box summarizing practice-oriented implications of our findings in Table 5.

## 5. Conclusions

This retrospective study demonstrates that MRI can detect early structural changes consistent with MRI-positive per JPS imaging criterion in patients with alcohol-related RAP. Three or more documented RAP episodes were identified as a significant risk factor for progression. The results support the role of repeated inflammatory injury as a driver of early ductal remodeling. Systematic post-discharge imaging during the inter-critical phase proved valuable in revealing subclinical changes. Alcohol and tobacco exposure further amplified the risk of structural progression. These findings suggest that closer surveillance is warranted in high-risk patients. Early recognition of ECP may open a window for targeted preventive strategies. Ultimately, timely intervention could help delay or mitigate the transition to established CP.

## Figures and Tables

**Figure 1 jcm-14-07830-f001:**
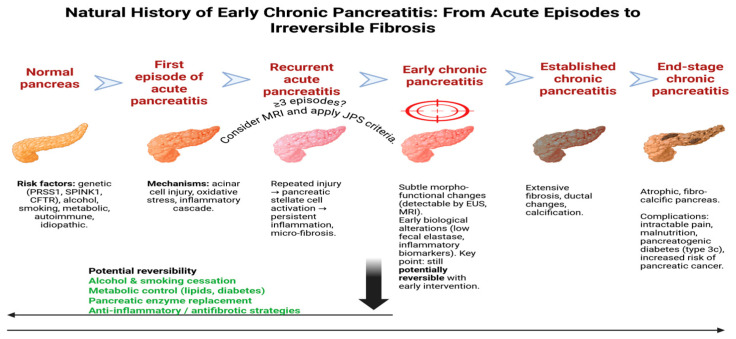
The natural history of early chronic pancreatitis: from acute episodes to irreversible fibrosis.

**Figure 2 jcm-14-07830-f002:**
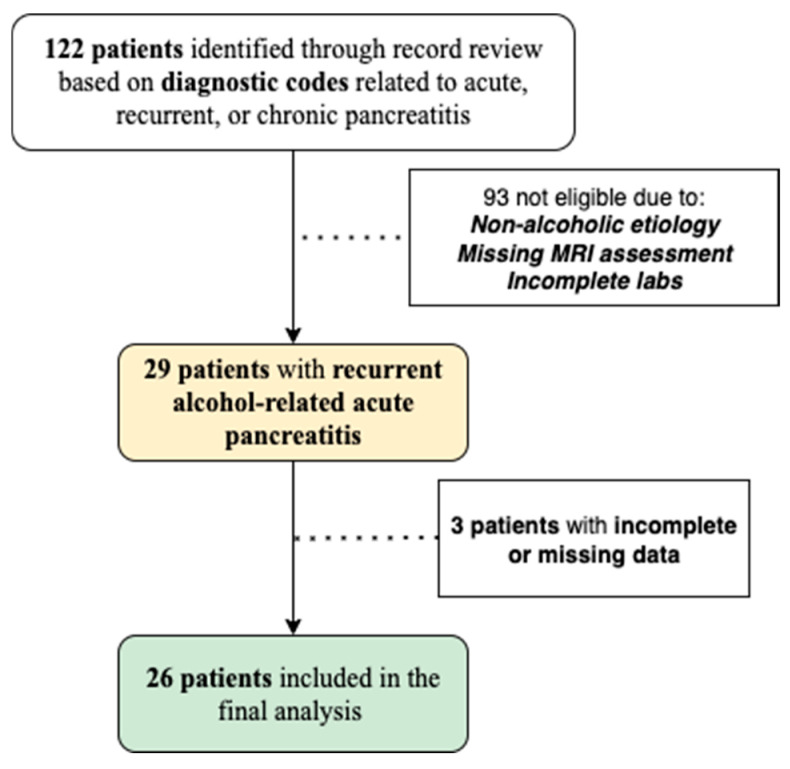
Flowchart detailing the selection process.

**Figure 3 jcm-14-07830-f003:**
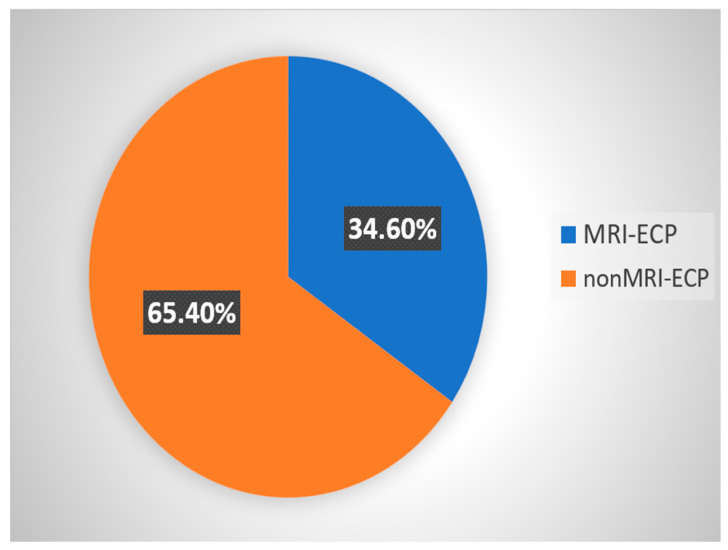
Patient classification.

**Table 1 jcm-14-07830-t001:** Baseline characteristics of the included patients.

Variable	Total (*n* = 26)	nonMRI-ECP (*n* = 17)	MRI-ECP (*n* = 9)	*p*-Value
Sex				0.043
Male	24 (92.3%)	17 (100%)	7 (77.8%)	
Female	2 (7.7%)	0 (0%)	2 (22.2%)	
Age (years, mean ± SD)	51.46 ± 9.31	50.25 ± 7.05	53.13 ± 11.03	0.025
Body weight status				–
Normal weight	19 (73.1%)	12 (70.6%)	7 (77.8%)	
Overweight	0 (0%)	0 (0%)	0 (0%)	
Obese	4 (15.4%)	4 (23.5%)	0 (0%)	
Underweight	3 (11.5%)	1 (5.9%)	2 (22.2%)	
Smoking status				
Active smokers	17 (65.4%)	10 (58.8%)	7 (77.8%)	0.418
Non-smokers	9 (34.6%)	7 (41.2%)	2 (22.2%)	–
Comorbidities				
Diabetes mellitus	8 (30.8%)	6 (35.3%)	2 (22.2%)	0.667
Arterial hypertension	6 (23.1%)	4 (23.5%)	2 (22.2%)	1.000

**Table 2 jcm-14-07830-t002:** Clinical parameters and disease progression between MRI-ECP and nonMRI-ECP groups.

Parameter	nonMRI-ECP (*n* = 17)	MRI-ECP (*n* = 9)	*p*-Value
Number of AP episodes	2.24 ± 0.56	3.89 ± 2.26	0.021
Disease duration (days)	4294 ± 1785	3547 ± 2565	0.034
Balthazar score	6.1 ± 2.3	5.5 ± 2.0	0.909
CTSI score	8.2 ± 3.1	5.9 ± 2.7	0.374

**Table 3 jcm-14-07830-t003:** Comparison of biochemical parameters.

Parameter	nonMRI-ECP (*n* = 17)	MRI-ECP (*n* = 9)	*p*-Value
Serum calcium	Normal: 16 (94.1%) Low: 1 (5.9%) High: 0 (0%)	Normal: 8 (88.9%) Low: 0 (0%) High: 1 (11.1%)	0.372
CRP level	Median rank: 14.76	Median rank: 11.11	0.246

**Table 4 jcm-14-07830-t004:** Multivariate analysis using penalized logistic regression.

Variable	OR	95% CI	*p*-Value
Number of AP episodes	4.00	0.79–20.3	0.09
Age (years)	1.52	0.96–2.40	0.07
Gender (female vs. male)	—	not significant	—
Episodes × Age	—	not significant	—

**Table 5 jcm-14-07830-t005:** Key Clinical Box–Practice-Oriented Insights.

Repeated alcohol-related acute pancreatitis episodes (≥3) represent a clinically relevant threshold associated with ECP.
The number of acute pancreatitis episodes was the only factor independently associated with ECP
MRI, applying the 2019 JPS criteria (>3 dilated side branches), provides superior sensitivity for early ductal alterations compared with CT.
Early recognition of ECP creates a potential window for preventive strategies and optimized surveillance pathways
Performing MRI systematically during the inter-critical period (4–8 weeks post-discharge) allows standardized and timely detection of subclinical disease.
CT-based severity scores (Balthazar, CTSI) did not discriminate between RAP and ECP in early stages
Alcohol and tobacco use remain major modifiable determinants of structural pancreatic progression.
The relatively high prevalence of MRI-defined ECP (34.6%) may underline the need for routine early imaging in alcohol-related RAP
Early recognition of ECP creates a potential window for preventive strategies and optimized surveillance pathways.

## Data Availability

All data supporting the findings of this study are available from the corresponding author upon reasonable request.

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
