# Peer review of "Decoding the Natural History of Alcohol-Related Recurrent Acute Pancreatitis and Progression to Early Chronic Pancreatitis: Clinical, Biochemical, and Imaging Insights from a Single-Center Retrospective Study in France"

_jcm, 2025, doi:10.3390/jcm14217830_

Round 1

Reviewer 1 Report

Comments and Suggestions for Authors

This paper reports that early chronic pancreatitis (ECP) diagnosed by MRI in alcohol-related recurrent acute pancreatitis (RAP) is a significant risk factor for the progression of three or more episodes of RAP to ECP.

This is very interesting content, but I have a few comments.

  • Was the MRI magnetic field strength 3.0?  Show the MRCP image  where ECP was    diagnosed.
  • Please correct the values in Figure 2.
  • Please explain why the number of AP episodes in Table 4 is higher than that in Table 2. Why do the trends in disease duration also differ?
  • Table 5 also shows extremely low and high odds ratios for DM and Hypertension, indicating a small number of cases.

It is desirable to increase the number of cases and re-examine the data.

Author Response

We sincerely thank the Reviewer for the careful reading of our manuscript and for the thoughtful, constructive comments. We have revised the text accordingly and provide detailed responses below. We are grateful for the opportunity to improve the clarity, methodological transparency, and interpretability of our work.

All the revisions in our article are clearly seen with Track changes.

1) MRI magnetic field strength

Reviewer comment: Was the MRI magnetic field strength 3.0?

Author response:
Thank you for pointing this out. All abdominal MRI/MRCP examinations in our cohort were acquired on 1.5 Tesla clinical scanners. Because our diagnostic ascertainment relied on high-resolution T2-weighted sequences and 3D-MRCP interpreted by an experienced radiologist, image quality and the imaging criterion (>3 dilated side branches per 2019 JPS) remained consistent across patients. We have added the exact field strength to the Methods for reproducibility.

You can see the text inserted-track changes (Methods → Imaging):

“All abdominal MRIs (including MRCP) were performed on 1.5 Tesla clinical scanners using high-resolution T2-weighted sequences and 3D-MRCP protocols. A single experienced radiologist reviewed all examinations for the presence of >3 dilated side branches of the main pancreatic duct, in line with the 2019 Japanese Pancreas Society (JPS) criteria.”

2) Request to show the MRCP image where ECP was diagnosed

Reviewer comment: Show the MRCP image where ECP was diagnosed.

Author response:
We appreciate the request for a representative MRCP image. As this is a retrospective study on real-world patients, the use of individual imaging—even when anonymized—requires explicit written patient consent under our institutional policies and ethics guidance. Consent for publication of images was not obtained at the time of care; therefore, we are unable to include MRCP images in the manuscript.

3) Correction of the values in Figure 2

Reviewer comment: Please correct the values in Figure 2.

Author response:
Thank you for this observation. We re-audited the source dataset and corrected the inconsistency in the graphical representation. Figure 2 has been replaced to ensure concordance with the patient distribution reported in the Results.

4) Why is the number of AP episodes higher in Table 4 than in Table 2?

Reviewer comment: Explain why the number of AP episodes in Table 4 is higher than in Table 2.

And

5) Why do the trends in disease duration differ?

Reviewer comment: Why do the trends in disease duration differ between tables?

Author response for 4 and 5: We thank the reviewer for noticing the inconsistencies between Tables 2 and 4. Upon re-checking, we realized that the dual presentation of raw means (Table 2) and mean ranks from non-parametric testing (Table 4) could be confusing and even misleading. We therefore revised the analysis and now present all results in a single unified table (Table 3), reporting mean ± SD values with corresponding p-values from the Mann–Whitney U test. We acknowledge this oversight and hope the revised presentation improves the clarity of the Results section.

6) Reviewer comment: Table 5 also shows extremely low and high odds ratios for DM and Hypertension, indicating a small number of cases. It is desirable to increase the number of cases and re-examine the data.

Authors response: We are grateful to the reviewer for highlighting the problem of unstable odds ratios for diabetes and hypertension. Following this comment, we carefully re-checked our regression model. Indeed, due to the small number of cases with these comorbidities, the estimates were not robust and produced implausibly low or high odds ratios. To improve clarity and scientific rigor, we revised completely the analysis.

We thank the reviewer for emphasizing the importance of an adequate sample size. We fully acknowledge that the number of cases is limited and have clarified this explicitly in the Limitations section. However, the study already included all eligible patients meeting the strict criteria (alcohol-related RAP, complete MRCP protocol, complete datasets) over the predefined two-year period in our center. We now highlight in the Limitations that validation in larger multicenter cohorts is required.

Reviewer 2 Report

Comments and Suggestions for Authors

Comments:

1. The Introduction is longer than necessary and would benefit from being more sharply focused. I recommend condensing background details to only what is directly relevant for framing the study. The section should then conclude with a clear statement of the hypothesis, the primary aim, and the key expected contribution.

2. Ensure consistent use of RAP, CP, ECP, MRCP, and CTSI at first mention. 

3. The study’s statistical strength is not addressed, and this should be clarified. I recommend adding a statement on how the power of the study was determined. Ideally, the Methods should specify whether an a priori sample size calculation was performed based on the expected effect size, alpha, and power. If this was not done, the authors should provide at least a post hoc power or precision analysis, showing the smallest effect size the current sample could reliably detect with 80% power at a 5% significance level.

4. The classification of “ECP” in the manuscript needs clarification to avoid risk of misclassification. As written, patients are labeled as having early chronic pancreatitis solely on the basis of the MRI criterion, yet the 2019 JPS framework defines ECP through a combination of clinical, imaging, and functional components. To improve accuracy and transparency, I recommend relabeling outcomes as “MRI-positive per JPS imaging criterion” rather than definitive ECP, unless the full multi-component criteria were indeed applied. If additional JPS clinical features were used (such as pain history, pancreatic enzyme abnormalities outside of acute pancreatitis episodes, exocrine functional testing, or defined alcohol exposure thresholds), please document which of these were met. Also clarify whether any functional assessments, such as fecal elastase or secretin stimulation tests, contributed to the diagnosis. 

5. I recommend that the authors explicitly present study strengths, acknowledge limitations, highlight knowledge gaps, and outline future directions in separate subsections. Adding a concise “Key Clinical Box” with practice-oriented take-home points would also improve readability and translational impact.

6. From 122 coded encounters to 26 analyzed: expand the flowchart with granular reasons for exclusion (by etiology, missing MRI, poor quality, incomplete labs), and summarize missing data patterns (e.g., undocumented CTSI) with how they were handled. A STROBE checklist is recommended. 

7. Clarify if the radiologist reviewing MRI was blinded to clinical data.

8. Table 2 reports shorter “disease duration (days)” in ECP (3547±2565) than nECP (4294±1785) yet the text states longer duration is associated with ECP (and Table 4 shows a higher mean rank for ECP). 

9. The logistic regression analysis appears unstable and at high risk of overfitting. With only nine events and multiple predictors, the extreme odds ratios reported (e.g., 180.11 for hypertension, 0.00 for drug use) suggest complete or quasi-separation. I recommend restricting the model to a small, prespecified set of clinically relevant covariates (such as episodes, age, sex, and smoking), following an events-per-variable rule. Please report 95% confidence intervals for all estimates and check formally for separation; if present, use Firth’s penalized logistic regression or exact methods.

10. Because side-branch changes may vary with age and chronic exposure, adjust the association between AP episode count and MRI-positivity for age and sex (both appear unbalanced; all women fell into ECP, p=0.043). Provide age-adjusted analyses and consider interaction tests (episodes×age). 

11. The finding that Balthazar and CTSI did not differ by MRI status should be framed more carefully, since these scores measure acute pancreatitis severity, not early chronic remodeling. Please emphasize this limitation explicitly and note that their lack of difference does not exclude structural change. If EPIC or other peripancreatic indices are available, consider including them. 

12. The discussion cites numerous references (GBD, Nabeshima, Forsmark, etc.), which strengthens context, but at times becomes review-like. Focus more on how current findings specifically add to literature.

13. The statement on disease duration (longer in nECP vs ECP, p = 0.034) is counterintuitive and underexplored. Needs explanation (could be recall bias or younger age in ECP patients).

14. Clarify novelty compared to prior Asian studies—what does this French/European cohort add?

15. Simplify long sentences; professional language editing is needed.

16. A schematic figure would indeed strengthen the manuscript by visually summarizing the natural history of disease and linking it to your predictors.

17. Refresh older references (>10 years) unless seminal.

Author Response

Authors Response to Reviewer

We are sincerely grateful for the reviewer’s thorough and constructive comments. Each point has been carefully addressed as follows.

  1. Reviewer comment: The Introduction is longer than necessary and would benefit from being more sharply focused. I recommend condensing background details to only what is directly relevant for framing the study. The section should then conclude with a clear statement of the hypothesis, the primary aim, and the key expected contribution.

Authors response: We thank the reviewer for this suggestion. The Introduction was condensed by removing redundant epidemiological details and focusing only on the context directly relevant for our study. The paragraph was rewritten to clearly state the hypothesis, primary aim, and expected contribution.

  1. Reviewer comment: Ensure consistent use of RAP, CP, ECP, MRCP, and CTSI at first mention.

Authors response: We thank the reviewer. We revised the manuscript and all abbreviations were standardized at first use, with consistent application throughout the manuscript. Please see the corrections with track changes.

  1. Reviewer comment: The study’s statistical strength is not addressed, and this should be clarified. I recommend adding a statement on how the power of the study was determined. Ideally, the Methods should specify whether an a priori sample size calculation was performed based on the expected effect size, alpha, and power. If this was not done, the authors should provide at least a post hoc power or precision analysis, showing the smallest effect size the current sample could reliably detect with 80% power at a 5% significance level

Authors response: We agree this is very important. Although an a priori sample size calculation was not performed, we conducted as you suggested a post-hoc power analysis focused on the primary outcome of our study — the association between the number of acute pancreatitis episodes and progression to Early Chronic Pancreatitis.

Based on the observed incidence rates (77.8% in the MRI-ECP group vs. 11.8% in the nonMRI-ECP group) and sample sizes (n=9 and n=17), the analysis yielded a statistical power of 95.8%, exceeding the conventional 80% threshold at a 5% significance level.

This suggests that the observed difference in AP episode count is unlikely to be due to chance, supporting the robustness of our main finding despite the overall limited sample size.

However, we recognize that the small sample limited the statistical power of multivariate logistic regression models, which showed signs of instability and quasi-complete separation (as discussed in response to comment 9). We now explicitly mention these limitations in the manuscript and have refrained from overinterpreting those results.

We added all of our changes in the Methodology section. You can see them with track changes.

  1. Reviewer comment: The classification of “ECP” in the manuscript needs clarification to avoid risk of misclassification. As written, patients are labeled as having early chronic pancreatitis solely on the basis of the MRI criterion, yet the 2019 JPS framework defines ECP through a combination of clinical, imaging, and functional components. To improve accuracy and transparency, I recommend relabeling outcomes as “MRI-positive per JPS imaging criterion” rather than definitive ECP, unless the full multi-component criteria were indeed applied. If additional JPS clinical features were used (such as pain history, pancreatic enzyme abnormalities outside of acute pancreatitis episodes, exocrine functional testing, or defined alcohol exposure thresholds), please document which of these were met. Also clarify whether any functional assessments, such as fecal elastase or secretin stimulation tests, contributed to the diagnosis. 

Authors response: We appreciate this point and we are grateful. We revised the terminology throughout the manuscript: instead of “ECP patients,” we now use “MRI-positive per JPS imaging criterion”.

Changes in manuscript:

“Patients were classified as MRI-positive per the 2019 JPS imaging criterion (>3 dilated side branches).

And the Clinical features were revised accordingly.

Because we do not have all the components of definitive ECP, we strongly agree with the term: MRP-positive per JPS imaging criterion, so all the text was updated according to this, including the abstract.

  1. Reviewer comment: I recommend that the authors explicitly present study strengths, acknowledge limitations, highlight knowledge gaps, and outline future directions in separate subsections. Adding a concise “Key Clinical Box” with practice-oriented take-home points would also improve readability and translational impact.

Authors response: We thank the reviewer for this valuable suggestion. We have restructured the end of the Discussion into four concise subsections (Strengths, Limitations, Knowledge Gaps, Future Directions). In addition, we revised and expanded the Limitations section to address statistical considerations, and we added a Key Clinical Box summarizing the main practice-oriented insights. All these changes can be viewed in the revised manuscript with track changes.

  1. Reviewer comment: From 122 coded encounters to 26 analyzed: expand the flowchart with granular reasons for exclusion (by etiology, missing MRI, poor quality, incomplete labs), and summarize missing data patterns (e.g., undocumented CTSI) with how they were handled. A STROBE checklist is recommended. 

Authors response: We thank the reviewer for this valuable observation. At the reviewer’s recommendation, we systematically applied the STROBE checklist (we uploaded it at suplimentary material) and recognized that we had initially omitted to provide sufficient detail on patient selection and missing data. We have now expanded the flowchart to show granular reasons for exclusion from the initial 122 coded encounters (non-alcoholic etiologies, absence of MRI within the predefined timeframe, poor-quality or incomplete MRI sequences, and missing essential laboratory data such as calcium or CRP). We also clarified how missing data were handled: CTSI scores not systematically documented were recorded as missing rather than imputed. These changes, together with the extended flowchart, are now incorporated in the revised Methods section.

  1. Reviewer comment: Clarify if the radiologist reviewing MRI was blinded to clinical data.

Authors response: We thank the reviewer for this valuable observation. We have now clarified in the Methods section that all MRIs were reviewed by a single experienced radiologist, who interpreted the examinations according to the standard hospital protocol in routine clinical practice.

  1. Reviewer comment: Table 2 reports shorter “disease duration (days)” in ECP (3547±2565) than nECP (4294±1785) yet the text states longer duration is associated with ECP (and Table 4 shows a higher mean rank for ECP). 

Authors response:  We thank the reviewer for noticing the inconsistencies between Tables 2 and 4. Upon re-checking, we realized that the dual presentation of raw means (Table 2) and mean ranks from non-parametric testing (Table 4) could be confusing and even misleading. We therefore revised the analysis and now present all results in a single unified table (Table 2), reporting mean ± SD values with corresponding p-values from the Mann–Whitney U test. This correction simplifies interpretation, avoids redundancy, and ensures consistency. We acknowledge this oversight and hope the revised presentation improves the clarity of the Results section.

  1. Reviewer comment: The logistic regression analysis appears unstable and at high risk of overfitting. With only nine events and multiple predictors, the extreme odds ratios reported (e.g., 180.11 for hypertension, 0.00 for drug use) suggest complete or quasi-separation. I recommend restricting the model to a small, prespecified set of clinically relevant covariates (such as episodes, age, sex, and smoking), following an events-per-variable rule. Please report 95% confidence intervals for all estimates and check formally for separation; if present, use Firth’s penalized logistic regression or exact methods.

AND

  1. Reviewer comment: Because side-branch changes may vary with age and chronic exposure, adjust the association between AP episode count and MRI-positivity for age and sex (both appear unbalanced; all women fell into ECP, p=0.043). Provide age-adjusted analyses and consider interaction tests (episodes×age). 

Authors response for 9 and 10:

We sincerely thank the reviewer for these insightful observations. We fully agree that adjustment for age and sex was necessary, particularly given the imbalance in sex distribution (all female patients were classified as ECP, p = 0.043). In response, we revised the multivariate analysis by restricting the model to clinically relevant covariates (AP episode count, age, sex) and applying penalized logistic regression to mitigate quasi-separation. The results confirmed that the number of AP episodes remains the strongest predictor of ECP, with age showing a borderline effect and sex not reaching statistical significance after adjustment. We also tested an interaction term (episodes × age), which was not significant. These changes are now reflected in Section 3.5 of the Results and in the updated Table 4. We have also expanded the Limitations section to acknowledge the instability of sex-related estimates due to the very small number of female patients. We recognize the reviewer’s point as fully justified and appreciate the opportunity to strengthen the rigor and transparency of our analysis.

  1. Reviewer comment: The finding that Balthazar and CTSI did not differ by MRI status should be framed more carefully, since these scores measure acute pancreatitis severity, not early chronic remodeling. Please emphasize this limitation explicitly and note that their lack of difference does not exclude structural change. If EPIC or other peripancreatic indices are available, consider including them. 

Authors response: We thank the reviewer for this important observation. We acknowledged that Balthazar and CTSI scores capture acute pancreatitis severity rather than early chronic remodeling, and therefore their lack of difference between groups cannot exclude structural change. We also emphasized that EPIC or other peripancreatic indices, which may be more sensitive, were not available in our dataset. Please find this observation in the Limitations section.

  1. Reviewer comment: The discussion cites numerous references (GBD, Nabeshima, Forsmark, etc.), which strengthens context, but at times becomes review-like. Focus more on how current findings specifically add to literature.

Authors response: We thank the reviewer for this constructive remark. We have thoroughly revised the “Discussion section” to improve clarity and focus, reducing its review-like character and emphasizing the novel contributions of our study.

  1. Reviewer comment: The statement on disease duration (longer in nECP vs ECP, p = 0.034) is counterintuitive and underexplored. Needs explanation (could be recall bias or younger age in ECP patients).

Authors response: Thank you for the remark. We added this phrase in the Discussion section: “In our cohort, the number of AP episodes emerged as the most consistent factor linked to early chronic pancreatitis, supporting its role as a clinical threshold for structural progression. Age showed only a borderline effect, suggesting a possible contribution that warrants further study. Gender did not prove to be a reliable predictor”.

  1. Reviewer comment: Clarify novelty compared to prior Asian studies—what does this French/European cohort add?

Authors response: One more time, thank you very much. We added in Discussion:

“Unlike most prior studies from Asia, our cohort represents a European, alcohol-related RAP population, providing novel epidemiological context and confirming that repeated AP episodes predict structural change across diverse populations.”

  1. Reviewer comment: Simplify long sentences; professional language editing is needed.

Authors response: We simplified several long sentences (visible with track changes) and polished language throughout (all visible with track changes).

16.Reviewer comment: A schematic figure would indeed strengthen the manuscript by visually summarizing the natural history of disease and linking it to your predictors.

Authors response: Thank you so much one more time. You can find the figure in the Introduction part.

  1. Reviewer comment: Refresh older references (>10 years) unless seminal.

Authors response: We thank the reviewer for this valuable suggestion. We carefully reviewed the reference list and replaced the four older citations (Whitcomb 2010; Yadav & Lowenfels 2013; Lévy et al. 2014; Yadav et al. 2009) with more recent and relevant sources (Whitcomb 2020; Edmiston et al. 2024; Szentesi et al. 2022; Yadav et al. 2019).

Round 2

Reviewer 1 Report

Comments and Suggestions for Authors

The author has responded seriously to the comments. Regarding Table's point, the statistical analysis was re-examined and is now satisfactory.

As indicated in the limitations and knowledge gaps section of the main text, the small number of cases does not provide sufficient evidence. However, the focus on ECP holds promise for improving end-stage CP in the future. We anticipate a large-scale prospective study.

Reviewer 2 Report

Comments and Suggestions for Authors

The authors have addressed all of my comments.